# Interaction and Metabolic Function of Microbiota during the Washed Processing of *Coffea arabica*

**DOI:** 10.3390/molecules28166092

**Published:** 2023-08-16

**Authors:** Xiaojing Shen, Baijuan Wang, Chengting Zi, Lulu Huang, Qi Wang, Chenchen Zhou, Wu Wen, Kunyi Liu, Wenjuan Yuan, Xingyu Li

**Affiliations:** 1College of Science, Yunnan Agricultural University, Kunming 650201, China; 2Research Center for Agricultural Chemistry, Yunnan Agricultural University, Kunming 650201, China; 3Yunnan Organic Tea Industry Intelligent Engineering Research Center, Kunming 650201, China; 4Key Laboratory of Intelligent Organic Tea Garden Construction in University of Yunnan Province, Kunming 650201, China; 5School of Wuliangye Technology and Food Engineering, Yibin Vocational and Technical College, Yibin 644003, China

**Keywords:** microorganism, metabolites, *Coffea arabica*, washed processing method, fermentation

## Abstract

Coffee fermentation is crucial for flavor and aroma, as microorganisms degrade mucilage and produce metabolites. This study aimed to provide a basis for understanding the impact of microorganisms on *Coffea arabica* from Yunnan, China, during washed processing. The microbial community structure and differentially changed metabolites (DCMs) of *C. arabica* beans during washed processing were analyzed. The results indicated that the top five predominant microorganisms at the genera level were *Achromobacter*, *Tatumella*, *Weissella*, *Streptococcus*, and *Trichocoleus* for bacteria and *Cystofilobasidium*, *Hanseniaspora*, *Lachancea*, *Wickerhamomyces*, and *Aspergillus* for fungi. Meanwhile, the relative content of 115 DCMs in 36 h samples decreased significantly, compared to non-fermentation coffee samples (VIP > 1, *p* < 0.05, FC < 0.65), and the relative content of 28 DCMs increased significantly (VIP > 1, *p* < 0.05, FC > 1.5). Furthermore, 17 DCMs showed a strong positive correlation with microorganisms, and 5 DCMs had a strong negative correlation (*p* < 0.05, |r| > 0.6). Therefore, the interaction and metabolic function of microbiota play a key role in the formation of coffee flavor, and these results help in clarifying the fermentation mechanisms of *C. arabica* and in controlling and improving the quality of coffee flavor.

## 1. Introduction

Coffee is the most popular and most widely consumed non-alcoholic beverage globally because of its unique flavor and taste. Mature coffee cherries are processed postharvest using different primary processing methods to obtain green coffee beans, which are roasted and ground to prepare coffee beverages. In this process, coffee fermentation is crucial in controlling coffee quality [1]. Microorganisms spontaneously participate in coffee fermentation and help remove coffee mucilage during coffee processing. These microorganisms also produce metabolites during fermentation, which can reach the interiors of coffee seeds, affecting coffee quality [2]. Therefore, the function of microorganisms has a crucial influence on coffee quality [3].

Yeast, bacteria, and filamentous fungi are the main microorganisms involved in coffee fermentation. Previous studies reported region-specific and method-specific microbiota characteristics during coffee fermentation [4]. Microbial species were richer in the washed processing method than in the other two coffee processing methods (natural and semi-dry processing), with the semi-dry processing method having minimal bacteria. In Brazil, yeasts, including *Pichia kluyveri*, *Candida glabrata*, and *Hanseniaspora opuntiae*, were found to be widely identified in spontaneous coffee fermentation [5]. In Australia, *Hanseniaspora uvarum*, *Pichia kudriavzevii*, *Leuconostoc mesenteroides*, and *Lactococcus lactis* were found to be the dominant microbiota during coffee fermentation [3]. Differences in microbial richness and diversity have also been reported in samples from a single country. For example, microbial richness and diversity in northern Colombia were found to be higher than they were in southwestern Colombia [6].

Improving coffee flavor quality through fermentation with selected special microbiota and different processing methods has become very popular and effective. Species of yeast—including *Picha fermentans*, *P. kudriavzevii*, *Saccharomyces cerevisiae*, and *Candida parapsilosis*—and bacteria— including *Lactiplantibacillus plantarum*, *Leuconostoc mesenteroids*, and *Bacilus subtilis*—have been used for coffee fermentation [7,8,9,10,11,12]. For example, coffee fermentation inoculation with *L. mesenteroides* improved the sensory score to 81, providing the coffee with dark chocolate, caramel, nutty, and spicy characteristics [7]. In addition, the cocultivation of *L. mesenteroides* with *Leuconostoc plantarum* led to a higher sensory score of 81.33 than that of a single inoculation with *L. mesenteroides* [7].

Yunnan province is the primary coffee plantation province in China, accounting for more than 95% of China’s coffee plantation area, making coffee one of the most important economic sources in the province. Zhang et al. studied the influence of processing conditions on coffee quality and metabolomic profiles in traditional wet-processing methods and reported that *Leuconostoc* and *Lactococcus* were active in *C. arabica* fermentation in China [13]. This study analyzed the microbiota and metabolites of coffee beans during washed processing to further study coffee quality and the functions of microbiota.

## 2. Results and Discussion

### 2.1. Microbial Community Structure of Coffee Based at the Phylum and Genus Level

Coffee has great microbiological diversity [1]. The sequences of bacteria and fungi ranged from 40,037 to 81,509 and from 38,944 to 179,398, respectively, with average lengths of 376 bp for bacteria and 241 bp for fungi. The coverage of coffee samples in each group was higher than 0.99. At the operational-taxonomic-unit (OTU) level, the ace index had *p*-values of 0.07 and 0.35 in bacteria and fungi, respectively (Figure 1A,E). The *p*-values of the Chaol index were 0.08 in bacteria and 0.35 in fungi. Among bacteria, WC0 had the highest Chaol index (Figure 1B), while among fungi, the highest Chaol index was observed in WC1 (Figure 1F). These results indicated that WC0 and WC1 had the highest species richness in bacteria and fungi, respectively.

The Shannon indices and the Simpon indices can reflect the number, evenness, and diversity of species of coffee samples [14]. The highest Shannon indices were observed for WC0 for bacteria (Figure 1C) and WC1 for fungi (Figure 1G), indicating that they had the highest species evenness for bacteria and fungi, respectively. On the other hand, WC3 showed the highest Simpon indices for bacteria and fungi (Figure 1D,H), suggesting that it had the highest observed diversity of both bacteria and fungi.

The bacteria in coffee samples from different fermentation times using the washed processing method were classified into 11 phyla by high-throughput sequencing, as shown in Figure 2A. These phyla included *Proteobacteria*, *Firmicutes*, *Actinobacteria*, *Cyanobacteria*, *Bacteroidota*, *Acidobacteriota*, *Gemmatimonadota*, *Chloroflexi*, and *Myxococcota*. Among them, the dominant phyla were *Proteobacteria* (comprising 54.36–61.60% of the community abundance at the phyla level), *Firmicutes* (8.57–41.25%), *Actinobacteria* (0.27–13.13%), *Cyanobacteria* (0.34–13.00%), and *Bacteroidota* (0.00–0.16%). Notably, the relative percentage of community abundance of *Proteobacteria* in all the samples was more than 50.00%. The relative percent of *Proteobacteria* community abundance initially increased from 57.52% in WC0 to 61.60% in WC2, then decreased to 54.36% in WC3. Moreover, *Firmicutes*, *Actinobacteria*, and *Cyanobacteria* exhibited significant changes with fermentation time lengthening. At the start of fermentation, WC0 showed a rich bacterial phyla composition, while at the end of fermentation, WC3 showed a more consistent bacterial phyla composition.

Furthermore, these bacteria were confirmed to 10 genera, as shown in Figure 2B. These genera included *Achromobacter*, *Tatumella*, *Weissella*, *Streptococcus*, *Trichocoleus*, *Burkholderia*, *Gluconobacter, Pseudenterobacter*, *Leuconostoc*, and *Dyella*. The top five genera were *Achromobacter, Tatumella, Weissella, Streptococcus, and Trichocoleus*. At the beginning of the fermentation, *Achromobacter* and *Trichocoleus* were the dominant bacteria, with 12.95 and 12.88 relative percentages of community abundance at the genus level, respectively. *Achromobacte* remained relatively stable during the first 12 h, but its abundance increased over the fermentation process to 24.69% at 24 h and 24.18% at 36 h. In contrast, *Trichocoleus* significantly declined as the fermentation processed, reaching its lowest abundance of 0.28% at 12 h. *Burkholderia* and *Dyella* showed changes similar to those of *Trichocoleus*. On the other hand, *Weissella* and *Leuconostoc* continuously increased throughout the fermentation process, from 0.15% to 33.74% and 0.00% to 6.39%, respectively. The population of *Tatumella* followed a similar pattern to that of *Streptococcus*, increasing during the first 12 h and then decreasing. The microbial community was highly diverse in coffee fermentation, especially in wet coffee fermentation [4]. Lactic acid bacteria species, especially *Leuconostoc*, were commonly involved in fermentation [4]. Low *Weissella* sp. and *Streptococcus faecalis* populations were also identified [4]. *Streptococcus* was the perdominant species during the fermentation in the wet process [4].

The fungi in different coffee samples were of lower diversity than bacteria, which could be classified into seven phyla by high-throughput sequencing, as shown in Figure 2C. These phyla were *Ascomycota*, *Basidiomycota*, *Mortierellomycota*, *Mucoromycota*, *Neocallimastigomycota*, *Olpidiomycota*, and *Rozellomycota*. Among them, the dominant phyla were *Ascomycota* (comprising 40.51–87.47% of the community abundance at phyla level) and *Basidiomycota* (11.80–59.30%). Notably, the relative abundance of *Ascomycota* in different samples was consistently higher than 40.00%. As the fermentation time increased, the relative abundance of *Ascomycota* first increased, reaching a maximum of 87.47% at 12 h (WC1), the decreased. On the other hand, the relative abundance of *Basidomycota* initially decreased, then increased, reaching a maximum of 59.30% at 36 h (WC3). *Ascomycota* is the most diverse and richest phylum in the kingdom of fungi, with 110,000 known species [15]. Due to their significant metabolic flexibility, *Ascomycota* members are widely used in various biotechnological applications, such as the production of fatty alcohols, fatty acids, and biofuels and the reduction and degradation of chemicals and solvents [15,16]. *Basidiomycota* is also a major phylum in the kingdom of fungi, with more than 31,000 known species, which is second only to *Ascomycota* in species numbers [17].

Furthermore, these fungi in coffee samples were confirmed to belong to 10 genera, as shown in Figure 2D. These genera were *Cystofilobasidum*, *Hanseniaspora*, *Aspergillus*, *Thermomyces*, *Lachances*, *Wickerhamomyces*, *Rasamsonia*, *Candida*, *Metschnikowia*, and *Apiotrichum*. The predominant genera were *Cystofilobasidu, Hanseniaspora, Lachancea, Wickerhamomyces*, and *Aspergillus*. The percentage of community abundance of *Cystofilobasidum* at the genus level ranged from 7.54% to 53.41%, with the maximum value observed at 36 h (WC3) and the minimum value at 12 h (WC2). On the other hand, the maximum value for *Aspergillus* was 10.76% at 12 h (WC1), and the minimum value was 0.90% at 36 h (WC3). *Wickerhamomyces* exhibited a similar pattern to that of *Aspergillus,* with a maximum of 7.93% at 12 h (WC1) and a minimum of 3.46% at 36 h (WC3). The percentages of community abundance of *Hanseniaspora* were 3.45% at 0 h (WC0) and 26.80% at 24 h (WC2). For *Lachancea*, the values of WC1 and WC2 were nearly equivalent at about 9%, and the minimum value was 3.38% at 36 h (WC3). The diversity of fungi in wet coffee processing is often lower than in dry and semi-dry processing, mainly because of the shorter fermentation time and the submerged environment [4]. The main filamentous fungi are *Aspergillus*, *Penicillium*, *Fusarium*, *Rhizopus*, *Mucor*, and *Cladosporium* [4,18]. Yeast species such as *Hanseniaspora* and *Candida* were frequently isolated from different locations worldwide [4].

Although many microorganism species are common in coffee fermentation, some are specific to certain regions. For example, in Australia, *Citrobacter* and *Leuconostoc* are predominant genera, along with *Pichia* and *Hanseniaspora* [3]. In Brazil, *Pichia*, *Candida*, and *Hanseniaspora* are the most frequent isolated genera [5]. In India, *Saccharomyces*, *Shizosaccharomyces*, *Bacillus*, *Lactobacillus*, *Leuconostoc*, *Pseudomonas*, and *Flavobacterium* are dominant genera during initial fermentation stages [18]. In China, *Enterobacter*, *Bacillus*, *Pseudomonas*, *Gluconobacter*, *Kluyvera*, and *Candida* are dominant in the wet processing of *C. arabica* [13]. Based on the results of this study, *Achromobacter, Tatumella, Weissella, Streptococcus*, and *Trichocoleus* were found to be predominant in the bacterial community, while *Cystofilobasidu, Hanseniaspora, Lachancea, Wickerhamomyces, and Aspergillus* were predominant in the fungal community during complete washed processing.

### 2.2. Differentially Changed Matabolities (DCMs) Analysis

A total of 2548 metabolites were detected in coffee samples from different fermentation times during washed coffee processing. These metabolites were classified into 17 super-classes, as shown in Figure 3. These super classes included lipids and lipid-like molecules (511 metabolites), organoheterocyclic compounds (439 metabolites), organic acids and derivatives (422 metabolites), organic oxygen compounds (332 metabolites), phenylpropanoids and polyketides (268 metabolites), benzenoids (233 metabolites), nucleosides, nucleotides, and analogs (69 metabolites), organic nitrogen compounds (31 metabolites), alkaloids and derivatives (30 metabolites), hydrocarbons (10 metabolites), lignans, neolignans, and related compounds (4 metabolites), hydrocarbon derivatives (2 metabolites), homogeneous non-metal compounds (2 metabolites), organic 1,3-dipolar compounds (1 metabolite), organohalogen compounds (1 metabolite), acetylides (1 metabolite), and others (192 metabolites).

Then, these metabolites were further grouped into 154 classes. These classes mainly included carboxylic acids and derivatives (352 metabolites), organooxygen compounds (331 metabolites), fatty acyls (203 metabolites), prenol lipids (157 metabolites), benzene and substituted derivatives (128 metabolites), flavonoids (88 metabolites), steroids and steroid derivatives (84 metabolites), phenols (60 metabolites), indoles and derivatives (54 metabolites), glycerophospholipids (52 metabolites), cinnamic acids and derivatives (49 metabolites), pyridines and derivatives (45 metabolites), coumarins and derivatives (41 metabolites), benzopyrans (37 metabolites), organonitrogen compounds (31 metabolites), isoflavonoids (27 metabolites), keto acids and derivatives (27 metabolites), imidazopyrimidines (26 metabolites), quinolines and derivatives (25 metabolites), lactones (19 metabolites), purine nucleosides (19 metabolites), hydroxy acids and derivatives (18 metabolites), naphthalenes (17 metabolites), phenylpropanoic acids (17 metabolites), pyrans (16 metabolites), pyrimidine nucleoside (16 metabolites), dihydrofurans (15 metabolites), piperidines (15 metabolites), peptidomimetics (13 metabolites), azoles (12 metabolites), diazines (12 metabolites), glycerolipids (10 metabolites), heteroaromatic compounds (10 metabolites), pyrrolidines (10 metabolites), and others.

The metabolites detected in different coffee samples were 2441 in WC0, 2391 in WC1, 2409 in WC2, and 2384 in WC3, respectively. There were 47 metabolites found exclusively in WC0, such as isomaltol, cimifugin, 4-methylumbelliferyl acetate, PE (15:0/22:1(13Z)), atalantoflavone, and luteoforol, which might be consumed during fermentation. There were 20 metabolites exclusively found in WC3, including salsoline-1-carboxylate, (S)-2,3-dihydro-7-hydroxy-2-methyl-4-oxo-4H-1-benzopyran-5-acetic acid, indole-3-acetaldehyde, capecitabine, sterigmatocystin, 8-methoxykynurenate, versetamide, calystegin A3, and amphetamine, which might be products of the fermentation process. Additionally, some metabolites, such as oleuropein, prosaikogenin A, and 2-amino-4-[(2-hydroxy-1-oxopropyl)amino]butanoic acid, were found in WC1, WC2, and WC3, indicating that they may have been produced during the initial stages and throughout the fermentation process.

Coffee contains a variety of chemical compounds, including nucleotides and derivatives, tannins, flavonoids, alkaloids, benzene and derivatives, phenylpropanoids, amino acid and derivatives, lipids, heterocyclic compounds, carboxylic acids and derivatives, saccharides, and others [19]. Members of these compounds have bioactive functions for health. Some of these compounds, such as sugar, proteins, amino acid, and phenolic compounds, are coffee flavor precursors, which form coffee aroma through Maillard reactions, Strecker degradation, caramelization reaction, and fragmentation reactions. [20]. For example, furanones, such as 4-hydroxy-2,5-dimethyl-3(2H)-furanone and 2(5)-ethyl-4-hydroxy-5(2)-methyl- 3(2H)-furan, generated by the Maillard reaction and subsequent aldol condensation, significantly influence the sweet caramel aroma of roasted coffee [21].

Caffeine, trigonelline, chlorogenic acids, carboxylic acids, carbohydrates and polymeric polysaccharides, lipids, protein, melanoidins, and minerals are crucial to coffee flavor [21,22]. Caffeine can contribute to coffee bitterness [21,22], while trigonelline contributes to its overall aroma by forming aromatic compounds [21,22]. Furthermore, four chlorogenic acids (chlorogenic acid, isochlorogenic acid, cryptochlorogenic acid, and trans-chlorogenic acid), four feruloylquinic acids (feruloylquinic acid, 3-feruloylquinic acid, 3-*O*-feruloylquinic acid, and 4,5-diferuloylquinic acid), and 14 caffeoylquinic acids (1-caffeoylquinic acid, 1-*O*-caffeoylquinic acid, 5-caffeoylquinic acid, cis-5-caffeoylquinic acid, dicaffeoylquinic acid, 1,5-dicaffeoylquinic acid, 3,5-di-*O*-caffeoylquinic acid, 3,4-di-O-caffeoylquinic acid, 1,3-dicaffeoylquinic acid, 4,5-dicaffeoylquinic acid, 3-caffeoyl-4-feruloylquinic acid, 4-*O*-caffeoyl-3-*O*-feruloylquinic acid, 3-*O*-caffeoyl-4-*O*-methylquinic acid, and 3,4,5-tricaffeoylquinic acid) were detected during wet processing, which undergo thermal degradation during calcination to form phenolic compounds and phenolic aromatic compounds that cause bitterness [22]. Citric acid and derivatives (methylisocitric acid, isocitric acid, and (1R,2R)-isocitric acid), malic acid and derivatives (2-isopropylmalic acid, and 3-isoproylmalic acid), chlorogenic acids and derivatives, quinic acid and derivatives (5-dehydroquinic acids, 3-dehydroquinic acid, and 5 coumaroylquinic acids (p-coumaroylquinic acid, 4-*O*-p-coumaroylquinic acid, 4-p-coumaroylquinic acid, 5-p-coumaroylquinic acid, and 3-*O*-p-coumaroylquinic acid, and 5-sinapoylquinic acid) were also detected. Citric, malic, chlorogenic, and quinic acids are the primary acidic compounds, occupying about 11% of green coffee beans, which form lactones and volatile phenols during the roasting of coffee beans, influencing coffee aroma [20,21].

When mature coffee cherries are harvested, a primary processing is necessary to obtain green coffee beans for the storage, transportation, and roasting of coffee [1]. There are various methods of obtaining green coffee beans in postharvest processing, including wet, dry, and semi-dry processing methods [1]. Wet processing is often used for *Coffee arabica* and involves submerged fermentation for 12–36 h after removing the mesocarp, resulting in coffee with a higher quality [23]. Fermentation is a natural and critical process for removing the mucilage and reducing water content [4,24] through enzymes that naturally occur in the coffee fruit and microflora acquired from the environment. During fermentation, microbial metabolites can migrate into the coffee and change various physiological parameters, such as water content, simple sugars, aroma, and other flavor precursors [4,25].

To gain further information about wet coffee processing, an overall comparison of the DCMs of WC0, WC1, WC2, and WC3 was carried out. As shown in Figure 4A, 82 DCMs were detected when comparing WC1 to WC0. These DCMs related to alkaloids and derivatives (one metabolite), benzenoids (four metabolites), lipids and lipid-like molecules (24 metabolites), nucleosides, nucleotides, and analogues (six metabolites), organic acids and derivatives (10 metabolites), organic nitrogen compounds (one metabolite), organic oxygen compounds (14 metabolites), organoheterocyclic compounds (12 metabolites), phenylpropanoids and polyketides (2 metabolites), and others (eight metabolites). Among them, the relative levels of 50 metabolites were decreased significantly (VIP > 1.0, *p* < 0.05, and FC < 0.67), included benzenoids (two metabolites, 4-hydroxy-3-methoxycinnmaldehyde, and diethyl phthalic acid), lipids and lipid-like molecules (16 metabolites, e.g., pimelic acid and carboprost methyl), nucleosides, nucleotides, and analogues (five metabolites, e.g., inosine, 5′-guanylic acid, and 3′-amino-3′-deoxythimidine glucuronide), organic acids and derivatives (seven metabolites, e.g., 4-fluoro-L-phenylalanine and γ-glutamylphenylalanine), organic oxygen compounds (six metabolites, e.g., hygromycin B and glucoheptonic acid), organoheterocyclic compounds (10 metabolites, e.g., isoeugenitol and guanine), phenylpropanoids and polyketides (one metabolite, 7-ethoxycoumarin), and others (three metabolites, securinine, sinapoyl aldehyde, and SM (d18:0/20:3(8Z,11Z,14Z)-2OH(5,6))). Among these DCMs, four metabolites, BULLEYACONITINE A, PS(15:0/24:1(15Z)), PC(18:2(9Z,12Z)/18:1(11Z)),and PC(14:1(9Z)/20:1(11Z)), were significantly decreased, with the value of FC less than 0.20. Meanwhile, the relative levels of 32 metabolites increased significantly (VIP > 1.0, *p* < 0.05, and FC > 1.5), including alkaloids and derivatives (one metabolite, harmalol), benzenoids (two metabolites, 4-(phenylamino)benzoic acid, and 3,4-dihydroxymandelic acid), lipids and lipid-like molecules (eight metabolites, e.g., 3-hexaprenyl-4-hydroxybenzoic acid, and pelanin), nucleosides, nucleotides, and analogues (one metabolite, uridine diphosphate glucose), organic acids and derivatives (three metabolites, 2-aminoisobutyric acid, portulacaxanthinⅡ, and 10-EdAM), organic nitrogen compounds (one metabolite, agmatine), organic oxygen compounds (eight metabolites, e.g., 3-hydroxynevirapine glucuronide and panthenol), organoheterocyclic compounds (two metabolites, niacinamide and N,Nitrosopiperidine), phenylpropanoids and polyketides (one metabolite, 4-feruloyl-1,5-quinolactone), and others (five metabolites, e.g., N-docosahexaenoyl lysine and 2,6-dihydroxynaphthalene). Among these DCMs, three metabolites, 3,4-dihydroxymandelic acid, portulacaxanthin II, and mycophenolic acid *O*-acyl-glucuronide were important increased with the value of FC over 5.0.

Similarly, 46 DCMs were detected between WC2 and WC1 (Figure 4B), including benzenoids (six metabolites), lipids and lipid-like molecules (11 metabolites), nucleosides, nucleotides, and analogues (two metabolites), organic acids and derivatives (nine metabolites), organic nitrogen compounds (three metabolites), organic oxygen compounds (six metabolites), organoheterocyclic compounds (five metabolites), phenylpropanoids and polyketides (two metabolites), and others (two metabolites). The relative levels of 28 DCMs were decreased significantly (VIP > 1.0, *p* < 0.05, and FC < 0.67), included benzenoids (five metabolites, e.g., menadione and isoeugenol), lipids and lipid-like molecules (eight metabolites, e.g., dodecanol and PE(16:0/0:0)), nucleosides, nucleotides, and analogs (one metabolite, inosine), organic acids and derivatives (three metabolites, 2-aminoisobutyric acid, 5-keto-D-gluconate, and enalaprilat), organic nitrogen compounds (two metabolites, betaine aldehyde and trolamine), organic oxygen compounds (three metabolites, caffeic acid 3-glucoside, D-gluconic acid, and difructose anhydrideⅢ), organoheterocyclic compounds (three metabolites, cantharidin, 7-methyladenine, and urocanic acid), phenylpropanoids and polyketides (two metabolites, 3-phenyllactic acid and trioxsalen), and others (one metabolite, DG(2:0/20:3(8Z,11Z.14Z)-2OH(5,6)/0:0)). Meanwhile, the relative levels of 18 DCMs were increased significantly (VIP > 1.0, *p* < 0.05, and FC > 1.5), including benzenoids (one metabolite, ginnalin B), lipids and lipid-like molecules (3 metabolites, palmitic amide, α-dimorphecolic acid, and stanozolol), nucleosides, nucleotides, and analogues (one metabolite, 5′-N-methylcarboxamidoadenosine), organic acids and derivatives (six metabolites, e.g., N-valylphenylalanine, and 5-hydroxy saxagliptin), organic nitrogen compounds (one metabolite, xestoaminol C), organic oxygen compounds (three metabolites, D-arabitol, α-CEHC glucuronide, and 2-deoxy-D-ribose), organoheterocyclic compounds (two metabolites, dihydrobiopterin and austdiol), and others (one metabolite, 6-bromoquinolin-2(1H)-one). Two DCMs, stanozolol and 2-deoxy-D-ribose, were significantly increased with a value of FC over 5.0, and 1 DCM, cantharidin, was significantly decreased with a value of FC less than 0.20.

A total of 45 DCMs were detected between WC3 and WC2 (Figure 4C), including alkaloids and derivatives (one metabolite), benzenoids (four metabolites), lipids and lipid-like molecules (eight metabolites), nucleosides, nucleotides, and analogues (1 metabolite), organic acids and derivatives (five metabolites), organic oxygen compounds (5 metabolites), organoheterocyclic compounds (11 metabolites), phenylpropanoids and polyketides (five metabolites), and others (five metabolites). Among them, the relative levels of 31 DCMs were decreased significantly (VIP > 1.0, *p* < 0.05, and FC < 0.67), including alkaloids and derivatives (one metabolite, mdo-npa), benzenoids (four metabolites, e.g.,bisnoryangonin, and phenyl acetate), lipids and lipid-like molecules (one metabolite, lysoPI(16:0/0:0)), nucleosides, nucleotides, and analogues (one metabolite, didanosine), organic acids and derivatives (five metabolites, e.g., tetrahydrodipicolinate and D-dopachrome), organic oxygen compounds (four metabolites, e.g., melilotoside and uridine 2′-phosphate), organoheterocyclic compounds (nine metabolites, e.g., indoleacetic acid and khellin), phenylpropanoids and polyketides (three metabolites, biochanin A, cis-*p*-coumaric acid, and biochanin A), and others (three metabolites, 13Z-docosenamide, citrinin hydrate, and PA(PGE2/i-16:0)). Meanwhile, the relative levels of 14 DCMs were increased significantly (VIP > 1.0, *p* < 0.05, and FC > 1.5), including lipids and lipid-like molecules (seven metabolites, e.g., 15-demethyl plumieride and digalactosyldiacylglycerol), organic oxygen compounds (one metabolite, (*S*)-Nerolidol 3-O-[a-L-Rhamnopyranosyl-(1->4)-a-L-rhamnopyranosyl-(1->2)-[4-(4-hydroxy-3-methoxycinnamoyl)-(E)-a-L-rhamnopyranosyl-(1->6)]-b-D-glucopyranoside]), organoheterocyclic compounds (two metabolites, lettucenin A, and 2,3-dimethylmaleic anhydride), phenylpropanoids and polyketides (two metabolites, 3″,6″-di-O-p-coumaroyltrifolin, and Fevicordin B 2-[rhamnosyl-(1->4)-glucosyl-(1->6)-glucoside]), and others (two metabolites, L-quinate, and DG(2:0/20:3(8Z,11Z,14Z)-2OH(5,6)/0:0)). Among them, L-quinate was a significantly increased DCM with a value of FC over 5.0 and bisnoryangonin was a significantly increased DCMs with a value of FC less than 0.20.

A total of 143 DCMs were detected between WC3 and WC0 (Figure 5), including benzenoids (20 metabolites), lipids and lipid-like molecules (31 metabolites), nucleosides, nucleotides, and analogues (eight metabolites), organic acids and derivatives (22 metabolites), organic nitrogen compounds (one metabolite), organic oxygen compounds (18 metabolites), organoheterocyclic compounds (29 metabolites), phenylpropanoids and polyketides (4 metabolites), and others (10 metabolites). Among them, the relative levels of 115 DCMs were decreased significantly (VIP > 1.0, *p* < 0.05, and FC < 0.67), including benzenoids (20 metabolites, e.g., ethyl benzoate and isoeugenol), lipids and lipid-like molecules (22 metabolites, e.g., 5-hexyl-2-furanoctanoic acid and valtrate), nucleosides, nucleotides, and analogues (six nVCs, e.g., inosine and guanosine), organic acids and derivatives (18 metabolites, e.g., trioxyethylene dimethacrylate and 3-methyl-2-oxovaleric acid), organic oxygen compounds (12 metabolites, e.g., melilotoside and D-gluconic acid), organoheterocyclic compounds (27 metabolites, e.g., isoeugenitol and cabotegravir), phenylpropanoids and polyketides (four metabolites, e.g., trioxsalen and sinapic acid), and others (six metabolites, e.g., citrinin hydrate and 2-aminophenol). Meanwhile, the relative levels of 28 DCMs were increased significantly (VIP > 1.0, *p* < 0.05, and FC > 1.5), including lipids and lipid-like molecules (nine metabolites, e.g., arctiol and α-dimorphecolic acid), nucleosides, nucleotides, and analogues (two metabolites, uridine diphosphate-N-acetylglucosamine, and uridine diphosphate glucose), organic acids and derivatives (four metabolites, e.g., sulbactam and portulacaxanthin Ⅱ), organic nitrogen compounds (one metabolite, xestoaminol C), organic oxygen compounds (six metabolites, e.g., aziridyl benzoquinone and panthenol), organoheterocyclic compounds (two metabolites, N-nitrosopiperidine, and niacinamide), and others (four metabolites, e.g., L-quinate and dodemorph). Meanwhile, 23 DCMs were important, with FC > 5.0 (DI1-DI3) or FC < 0.20 (DD1-DD20).

### 2.3. Correlation between Microorganisms and DCMs

Coffee beans contain rich precursors that can generate flavor and aroma compounds during roasting. Coffee fermentation can increase the diversity of coffee aroma and flavor compounds. Microorganisms in coffee not only produce important enzymes (such as pectin lyase, polygalacturonase, and pectin methyl esterase) for degrading pectin substances, but also produce diverse metabolites during coffee fermentation [25]. Therefore, microorganisms play a crucial role in coffee fermentation [25].

A correlation analysis between microorganisms and 23 DCMs was carried out to obtain more useful information about the function of microbes in the fermentation of coffee. Figure 6 presents a network that intuitively clarified the complex relationship between DCMs, bacteria, and fungi, in which correlation is represented by lines of different colors and thicknesses, while DCMs and microbes are represented by different shapes. A red line represents a positive correlation, while blue signifies a negative correlation. A darker color indicates that the correlation is stronger. Rectangles represent DCMs, circles represent bacteria, and triangles represent fungi. Based on the value of the correlation coefficient, 0.8–1.0 indicated an extremely strong correlation, 0.6–0.8 indicated a strong correlation, 0.4–0.6 indicated a moderate correlation, 0.2–0.4 indicated a weak correlation, and 0.0–0.2 indicated a weak correlation or no correlation [26]. *Metschnikowia* and *Apiotrichum* fungi genera were extremely strongly positively correlated with *Leuconostoc* (r > 0.8). Furthermore, 17 DCMs exhibited a strong positive correlation with microorganisms (0.8 > r > 0.6). Among them, 12 DCMs demonstrated a strong positive correlation with *Candida,* and four DCMs showed a strong positive correlation with *Burkholderia.* L-quinate showed a strong positive correlation with *Leuconostoc*, *Metschnikowia*, and *Apiotrichum*. Additionally, 4-hydroxy-6- methyl-3-(1-oxobutyl)-2H-pyran-2-one displayed a strong positive correlation with *Burkholderia*, *Dyella*, and *Candida*, while 3-deazaadenosine showed a strong positive correlation with *Burkholderia*, *Dyella*, *Wickerhamomyces*, and *Candida*. On the other hand, 5 DCMs exhibited a strong negative correlation with microorganisms (−0.6 > r > −0.8). Specifically, three DCMs (PC)14:1(9Z)/20:1911Z), PC(18:2(9Z,12Z)/18:1(11Z)), and PE-NMe(22:2(13Z,16Z)/16:1(9Z)) showed a strong negative correlation with *Hanseniaspora.* Additionally, portulacaxanthin Ⅱ had a strong negative correlation with *Candiada*, and 3-deazaadenosine showed a strong negative correlation with *Tatumella*. In addition, all 23 DCMs exhibited a moderate correlation with microorganisms. Altogether, microorganisms demonstrated an important impact in shaping coffee compounds during the washed processing.

## 3. Materials and Methods

### 3.1. Materials and Chemical Standards

Mature coffee cherries (*C. arabica*) were obtained from Puer City, Yunnan Province, China. After harvesting, the coffee cherries were processed using the washed processing method [4,20,23]. Initially, red mature coffee cherries were hand-picked and de-pulped to remove the skin and the pulp. Subsequently, the thin mucilaginous layer surrounding the coffee seeds was removed via fermentation. During the fermentation, four coffee bean samples were obtained at 0 h (WC0), 12 h (WC1), 24 h (WC2), and 36 h (WC3) (*n* = 3), respectively. The coffee bean samples were divided into two parts: one part was stored at −80 °C for high-throughput sequencing analysis, and the coffee beans were obtained by removing the parchment skin from the other part and stored separately at −20 °C for the analysis of metabolites. High-performance liquid chromatography (HPLC) grade methyl alcohol, acetonitrile, and propyl alcohol were purchased from Fisher Co., Ltd. (Shanghai, China). The MagAtrract PowerSoil Pro DNA Kit was purchased from Oiagen (Hilden, Germany).

### 3.2. High-Throughput Sequencing Analysis

High-throughput sequencing analysis of coffee samples during fermentation processing was carried out following DNA extraction and PCR amplification by Majorbio Bio-Pharm Technology Co. Ltd. (Shanghai, China). Microbial DNA was extracted following the MagAtrract PowerSoil Pro DNA Kit operation instructions. The absolute quantification was determined by 1.0% agarose gel electrophoresis. A total of 12 different spike-in sequences with four different concentrations (10^3^, 10^4^, 10^5^, and 10^6^ copies of internal standards) were added to the sample DNA pool. For bacteria, the hypervariable region V5-V7 of the 16S rRNA gene was amplified with forward primer 799F (AACMGGATTAAGATACCCKG) and reverse primer 1193R (ACGTCATCCCCACCTTCC). Meanwhile, the ITS1 region used ITS1F (CTTGGTCATTTAGAGGAAGTAA) and ITS2R (GCTGCGTTCTTCATCGATGC) primers for fungi [27]. The PCR conditions consisted of an initial step at 95 °C for 3 min, followed by 27 cycles (799F-1392R), 13 cycles (799F-1193R), and 35 cycles (ITS1F-ITS2R) of denaturation at 95 °C for 30 s, annealing at 55 °C for 30 s, and extension at 72 °C for 45 s. A final extension was conducted at 72 °C for 10 min. Raw FASTQ files were de-multiplexed using an in-house Perl script and, then, quality-filtered by fast version 0.19.6 [28] and merged by FLASH version 1.2.7 [29], based on the following criteria: the truncated reads over a 50 bp sliding window and an average quality score of <20; overlapping sequences longer than 10 bp and the maximum mismatch ratio of overlap region 0.2. Then, the optimized sequences were clustered into operational taxonomic units (OTUs) using UPARSE 7.1 with a 97% sequence similarity level [30,31]. The OTUs assigned to spike-in sequences were filtered out, and reads were counted. A standard curve (based on read counts versus spike-in DNA copy numbers) for each sample was generated, and the quantitative abundance of each OTU in a sample was determined. The taxonomy of each OTU representative sequence was analyzed by RDP Classifier version 2.2 [32] against the 16S rRNA gene database, using a confidence threshold of 0.7, then adjusted on the basis of the estimated rRNA operon copy number [33].

### 3.3. Metabolites Analysis

Metabolites of coffee beans during fermentation processing were extracted and analyzed using the liquid chromatography-mass-spectrometry (LC-MS/MS)-based metabolomics approach by Majorbio Bio-Pharm Technology Co. Ltd. (Shanghai, China). The coffee powder (CP) samples (50 mg) were accurately weighed and extracted using 0.4 mL 80% methanol solution with 0.02 mg/mL L-2-chlorophenylalanin as the internal standard. Quality control (QC) samples were prepared by mixing equal volumes of all samples to monitor the stability of the analysis [34].

Samples were injected into a UHPLC-Q Exactive system of Thermo Fisher Scientific for LC-MS analysis [35]. LC-MS data were preprocessed by Progensis QI software 3.0 (Waters Corporation, Milford, MA, USA) [36]. At the same time, the metabolites were searched and identified by the HMDB, Metlin, and Majorbio Database [37]. The response intensity of the sample mass spectrum peaks was normalized by the sum normalization method, and variables with relative standard deviation (RSD) > 30% of QC samples were removed, and log10 logarithmization was performed.

### 3.4. Statistical Analysis

Statistical analyses were performed using IBM SPSS Statistics 26.0 (SPSS Inc., Chicago, IL, USA). Variable importance in projection (VIP) analysis ranked the overall contribution of each variable to the OPLS-DA model, and those variables with VIP > 1.0, *p* < 0.05, and fold change (FC) > 1.5 or < 0.67 were classified as differentially changed metabolites (DCMs).

### 3.5. Accession Numbers

The raw sequencing reads of bacterial 16S rRNA and fungal ITS1 were deposited into the NCBI Sequence Read Archive (SRA) database (Accession Number: PRJNA992233 and PRJNA992238).

## 4. Conclusions

The microbial community structure and DCMs of *C. arabica* beans from Yunnan province in the coffee fermentation process using the washed process were compared. The bacterial community consisted of 11 phyla and 11 genera, while the fungi community had eight phyla and 11 genera. The predominant phyla were *Proteobacteria*, *Firmicutes*, *Actinobacteria*, *Cyanobacteria*, and *Bacteroidota* for bacteria and *Basidiomycota*, *Ascomycota*, *Mortierellomycota*, and *Mucoromycota* for fungi. The top five predominant bacterial genera were *Achromobacter*, *Tatumella*, *Weissella*, *Streptococcus*, and *Trichocoleus*, while the dominant fungal genera were *Cystofilobasidium, Hanseniaspora*, *Lachancea*, *Wickerhamomyces*, and *Aspergillus*. A total of 2548 metabolites were identified during the coffee fermentation process, which were classified into 17 classes. Furthermore, a total of 82 DCMs were detected in WC1/WC0. Among them, 50 DCMs decreased significantly and 32 DCMs increased. In WC2/WC1, 46 DCMs were detected, of which 28 DCMs decreased significantly, while 17 DCMs increased significantly. In WC3/WC2, 45 DCMs were detected, with 31 DCMs significantly decreased and 12 DCMs significantly increased. Overall, 143 DCMs were identified in WC3/WC0, with 115 DCMs decreased significantly and 28 DCMs increased significantly. The correlation analysis revealed that 17 DCMs had a strong positive correlation with microorganisms, while 5 DCMs had a strong negative correlation. These findings indicated that microorganisms have an important influence on coffee compounds during coffee fermentation. Further research is necessary to identify the crucial microbial genera that affect the coffee compounds, to understand the response and succession rules of these microorganisms during the coffee fermentation, and to determine their effects on the aroma and quality of coffee. Thus, obtaining key microorganisms that impact coffee flavor is a way to improve the quality of coffee flavor.

## Figures and Tables

**Figure 1 molecules-28-06092-f001:**
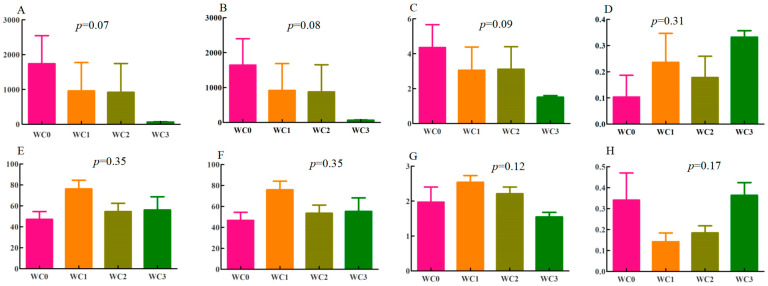
Analysis of microbial alpha diversity in wash processing during the fermentation process. ((**A**–**D**) Ace, Chaol, Shannon, and Simpson in bacteria, respectively; (**E**–**H**) Ace, Chaol, Shannon, and Simpson in fungi, respectively).

**Figure 2 molecules-28-06092-f002:**
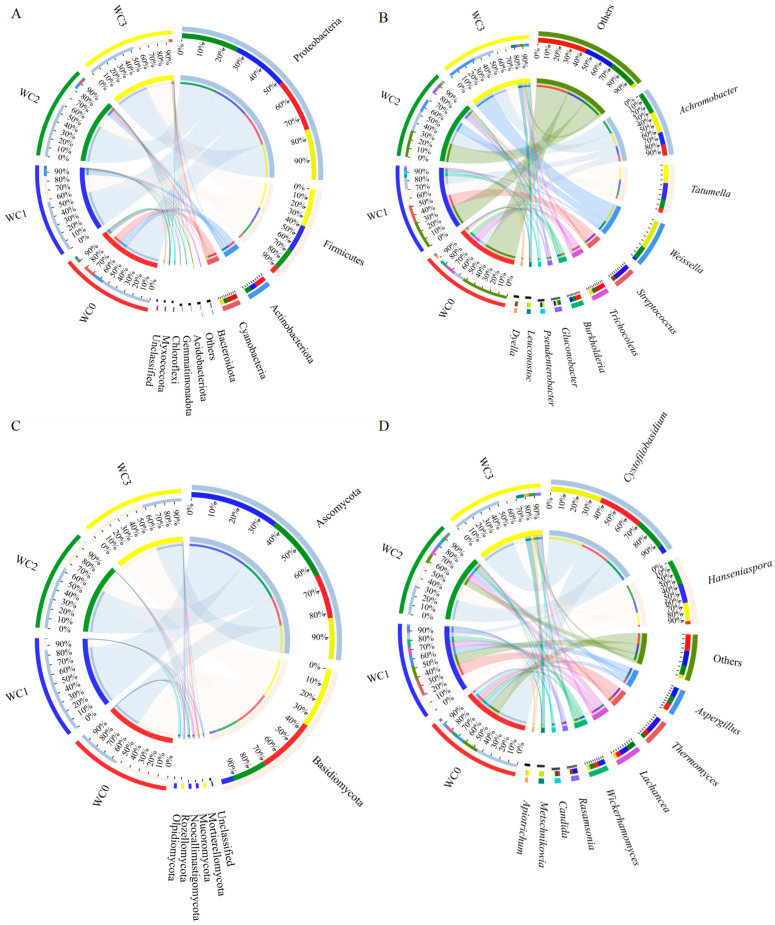
Microbial community structure in wash processing during the fermentation process. The percentage of community abundance on the phylum level of bacteria (**A**); the percentage of community abundance on the genus level of bacteria (**B**); the percentage of community abundance on the phylum level of fungi (**C**); the percentage of community abundance on genus level of fungi (**D**). Left: half circles mean the microbial community structure of different fermentation times, and different colors of left outer circles represent different coffee samples from fermentation times; among them, red represents WC0, blue represents WC1, green represents WC2, and yellow represents WC3, and the numbers represent the percentage of community abundance of different microorganism in these samples. Right: half circles mean the percentage of community abundance of different microorganisms on phylum or genus level, and different colors represent different microorganisms on phylum or genus level; the number represents the percentage of community abundance of these microorganisms in different samples.

**Figure 3 molecules-28-06092-f003:**
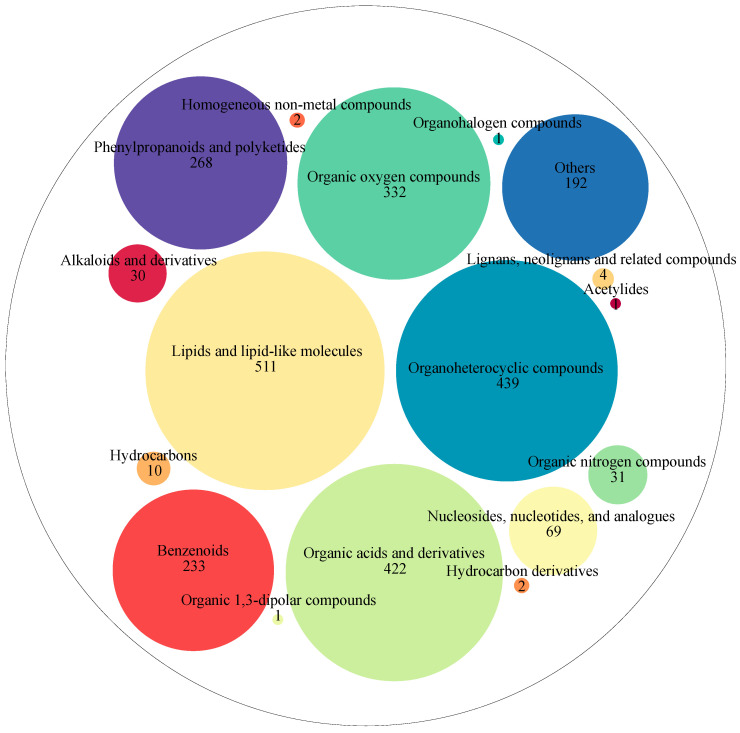
Super-classes of metabolites from different times of coffee fermentation by the washed primary processing method. Different colored circles represent different super-classes of metabolites, the different sizes of circles represent the numbers of metabolites, and the bigger the circle the greater the number of metabolites; the numbers represent the numbers of metabolites.

**Figure 4 molecules-28-06092-f004:**
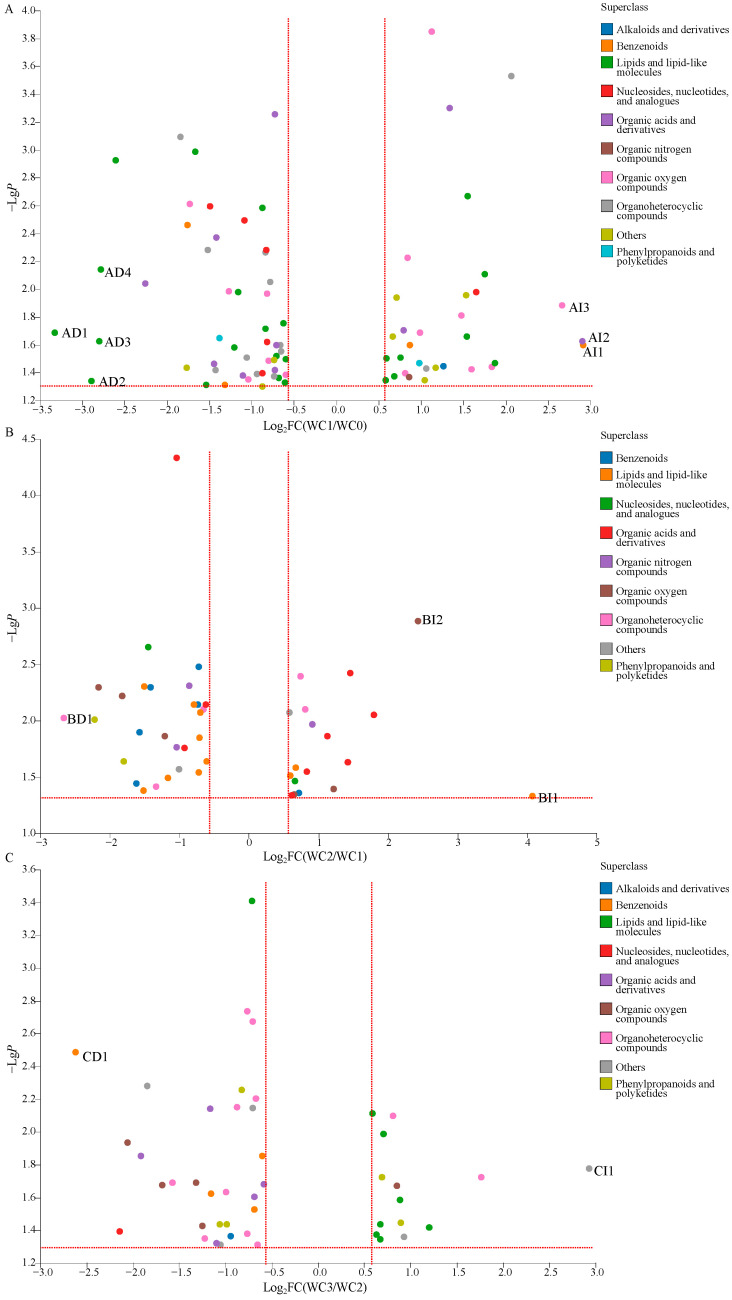
The DCMs between WC1 and WC0 (**A**), AI1: 3,4-dihydroxymandelic acid, AI2: portulacaxanthin II, AI3: mycophenolic acid *O*-acyl-glucuronide (FC > 5.0); AD1: BULLEYACONITINE A, AD2: PS(15:0/24:1(15Z)), AD3: PC(18:2(9Z,12Z)/18:1(11Z)), AD4: PC(14:1(9Z)/20:1(11Z)) (FC < 0.20); DCMs between WC2 and WC1 (**B**), BI1:stanozolol, BI2: 2-deoxy-D-ribose(FC > 5.0), BD1: cantharidin (FC < 0.20); DCMs between WC3 and WC2 (**C**), CI1: L-quinate (FC > 5.0), CD1: bisnoryangonin (FC < 0.20).

**Figure 5 molecules-28-06092-f005:**
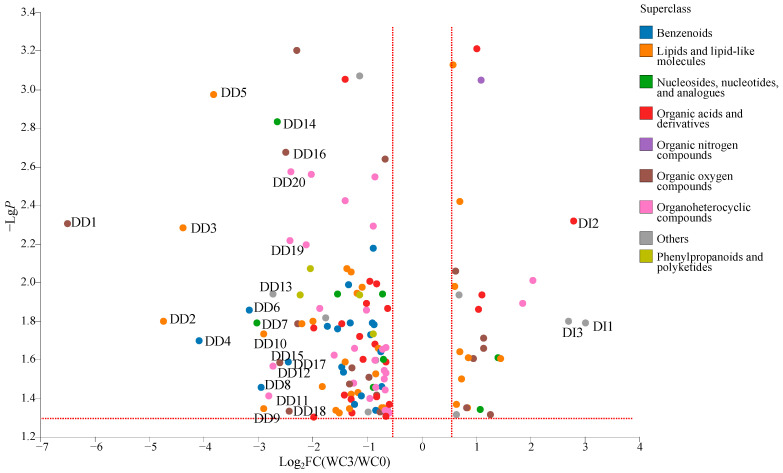
The DCMs between WC3 and WC0. (DI1: L-quinate, DI2: portulacaxanthin II, DI3: PG(LTE4/i-22:0) (FC > 5.0); DD1: glucoheptonic acid, DD2: PC(18:2(9Z,12Z)/18:1(11Z)), DD3: PC(14:1(9Z)/20:1(11Z)), DD4: coniferyl acetate, DD5: PE-NMe(22:2(13Z,16Z)/16:1(9Z)), DD6: diethyl phthalic acid, DD7: 3-deazaadenosine, DD8: ethyl benzoate, DD9: PS(15:0/24:1(15Z)), DD10: 9,10-epoxystearic acid, DD11: noradrenochrome, DD12: elenolide, DD13: SM(d18:0/20:3(8Z,11Z,14Z)-2OH(5,6)), DD14: Inosine, DD15: 4-Hydroxy-6-methyl-3-(1-oxobutyl)-2H-pyran-2-one, DD16: Uridine 2′-phosphate, DD17: Isoeugenol, DD18: Ethyl beta-D-glucopyranoside, DD19: Cabotegravir, DD20: 4(1H)-Pyridinone, 3-hydroxy-1-(3-hydroxypropyl)-2-methyl- (FC < 0.2).

**Figure 6 molecules-28-06092-f006:**
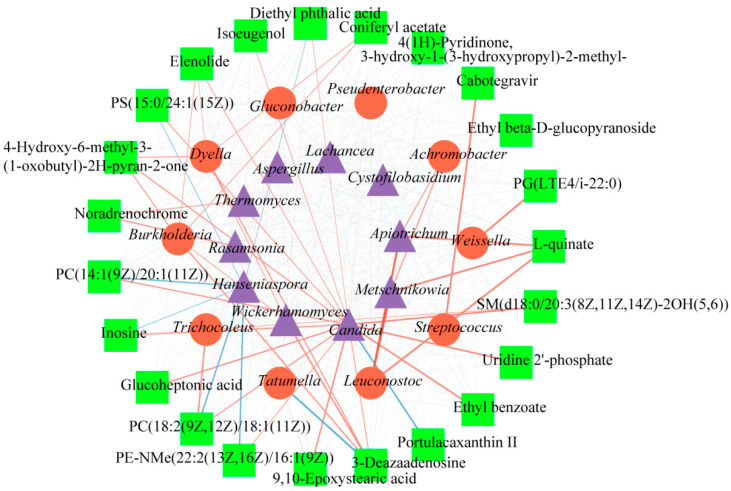
The interaction of microbes and significant DCMs in different fermentation times in coffee samples based on the Spearman correlation analysis (*p* < 0.05). Red solid lines mean positive correlation, solid blue lines mean negative correlation, thicker solid lines mean stronger correlation, and thinner solid lines mean weaker correlation.

## Data Availability

The experimental data provided in this work are available in the articles.

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
