# Peer review of "Interaction and Metabolic Function of Microbiota during the Washed Processing of Coffea arabica"

_molecules, 2023, doi:10.3390/molecules28166092_

Round 1

Reviewer 1 Report

The manuscript by Shen et all. presents the analysis of bacterial communities and their metabolites in fermented coffee samples. These results are of potential interest to a broad audience, specifically those involved in food science and coffee fermentation analysis. The paper is generally well-structured and but it needs a lot of improvements. First of all, the English usage in the paper needs improvement. I strongly suggest to get editing help from someone with full professional proficiency in English. Moreover:

1.        Italicize the Latin bacteria names regardless of the taxonomic level

2.       Line 16 – remove the word repetition “produce”

3.        The abstract section, sentence “Therefore, obtaining key microorganisms that impacted coffee flavor is a way to improve coffee flavor quality” needs development.  

4.       Line 55-59 – please add the reference

5.       Materials and methods section – there is no details about the washed processing method and sampling method, number of replicates etc. Please improve it.

6.       Line 82 -83 “The absolute quantification was determined by spike-in sequences that worked like internal standards.” – I do not understand it. Please briefly explain.

7.       Line 83 – which region of 16s rRNA gene was amplified? Please specify it. If you did not design the primers on your own please add the references.

8.       There is no details about sequencing analysis, please add appropriate paragraph.

9.       Line 95 “(e.g. Silva v138)”, Remove e.g.

10.   Line 127 – what is OUT, did you mean OTU?

11.   Figures 4 and 5 are unreadable. Please increase the font of the legend.

12.   Conclusions section: “Obtaining key microorganisms that impact coffee flavor is a way to 455

13.   improve coffee flavor quality.” – please expand it, it seems to be an interesting idea.

The manuscript by Shen et all. presents the analysis of bacterial communities and their metabolites in fermented coffee samples. These results are of potential interest to a broad audience, specifically those involved in food science and coffee fermentation analysis. The paper is generally well-structured and but it needs a lot of improvements. First of all, the English usage in the paper needs improvement. I strongly suggest to get editing help from someone with full professional proficiency in English. Moreover:

1.        Italicize the Latin bacteria names regardless of the taxonomic level

2.       Line 16 – remove the word repetition “produce”

3.        The abstract section, sentence “Therefore, obtaining key microorganisms that impacted coffee flavor is a way to improve coffee flavor quality” needs development.  

4.       Line 55-59 – please add the reference

5.       Materials and methods section – there is no details about the washed processing method and sampling method, number of replicates etc. Please improve it.

6.       Line 82 -83 “The absolute quantification was determined by spike-in sequences that worked like internal standards.” – I do not understand it. Please briefly explain.

7.       Line 83 – which region of 16s rRNA gene was amplified? Please specify it. If you did not design the primers on your own please add the references.

8.       There is no details about sequencing analysis, please add appropriate paragraph.

9.       Line 95 “(e.g. Silva v138)”, Remove e.g.

10.   Line 127 – what is OUT, did you mean OTU?

11.   Figures 4 and 5 are unreadable. Please increase the font of the legend.

12.   Conclusions section: “Obtaining key microorganisms that impact coffee flavor is a way to 455

13.   improve coffee flavor quality.” – please expand it, it seems to be an interesting idea.

Author Response

  1. Comment: “The manuscript by Shen et all. presents the analysis of bacterial communities and their metabolites in fermented coffee samples. These results are of potential interest to a broad audience, specifically those involved in food science and coffee fermentation analysis. The paper is generally well-structured and but it needs a lot of improvements. First of all, the English usage in the paper needs improvement. I strongly suggest to get editing help from someone with full professional proficiency in English. Moreover:

Response: Thank you the comment from reviewer 1. These positive suggestions

about this manuscript are very inspiring for us. And it has been edited by a full professional proficiency in English.

  1. Comment: “Italicize the Latin bacteria names regardless of the taxonomic level”.

Response: This suggestion is very important and all Latin bacteria names have been revised using Italicize.

  1. Comment: “Line 16 – remove the word repetition “produce”.

Response: This suggestion is very important and it has been revised.

  1. Comment: “The abstract section, sentence “Therefore, obtaining key microorganisms that impacted coffee flavor is a way to improve coffee flavor quality” needs development.”

Response: This suggestion is very important and it has been revised.

  1. Comment: “Line 55-59 – please add the reference”

Response: This suggestion is very important and it has been revised.

  1. Comment: “Materials and methods section – there is no details about the washed processing method and sampling method, number of replicates etc. Please improve it.”

Response: This suggestion is very important and it has been revised.

  1. Comment: “Line 82 -83 “The absolute quantification was determined by spike-in sequences that worked like internal standards.” – I do not understand it. Please briefly explain.”

Response: This suggestion is very importan and the method of absolute quantification was detailed described as following: the quality and concentration of DNA were determined by 1.0% agarose gel electrophoresis and a NanoDrop2000 spectrophotometer (Thermo Scientific, United States). 12 different spike-in sequences with four different concentrations (103, 104, 105 and 106 copies of internal standards) were added to the sample DNA pools. Spike-in sequences consisted of conserved regions identical to those of selected natural 16S rRNA genes and artificial variable regions shared negligible identity nucleotide sequences with the public databases, which worked like internal standard and facilitated the absolute quantification across samples. Additionally, we have revised this section.

  1. Comment: “Line 83 – which region of 16s rRNA gene was amplified? Please specify it. If you did not design the primers on your own please add the references.”

Response: This suggestion is very important and it has been revised.

  1. Comment: “There is no details about sequencing analysis, please add appropriate paragraph.”

Response: This suggestion is very important and it has been revised.

  1. Comment: “Line 95 “(e.g. Silva v138)”, Remove e.g.”

Response: This suggestion is very important and it has been revised.

  1. Comment: “Line 127 – what is OUT, did you mean OTU?”

Response: This suggestion is very important and it has been revised.

  1. Comment: “Figures 4 and 5 are unreadable. Please increase the font of the legend.”

Response: This suggestion is very important and it has been revised.

  1. Comment: “Conclusions section: “Obtaining key microorganisms that impact coffee flavor is a way to improve coffee flavor quality.” – please expand it, it seems to be an interesting idea.”

Response: This suggestion is very important and it has been revised.

Reviewer 2 Report

1. The authors study the different fermentation time on the changing of composition of microbiota and metabolites and their relationships. It’s an interesting topic to understanding the roles of the inherent microbiota on metabolites. The authors use NGS and LC-MS analysis to get abundant information, but analysis among different fermentation time points are not done.

2. There are many mistyping and misspelling in the content. Extensive editing of English language required.

3. In materials and methods, all the processing steps of the coffee bean after harvest and before analysis should be described in detail.

4. Sample codes describe in the content and in the figure legends are different, please unify.

5. The legend of Figure 2 should be improved. What's the meaning of inner and outer circles and different colors? And the meaning of the number and percentage beside the circles. The same as Figure 3. What are the meaning of different circles of the same color and different size of circles, And the open circle in upper right side.

6. Are DCM and DCnVCs mean the same? If yes, please unify; if not, please check the content and correct.

7. Almost no discussion on the results and what are the meaning and importance of your finding.

8. Conclusion should be made by summarizing the results briefly, not repeat the results again.

Extensive editing of English language required.

Author Response

  1. Comment: “The authors study the different fermentation time on the changing of composition of microbiota and metabolites and their relationships. It’s an interesting topic to understanding the roles of the inherent microbiota on metabolites. The authors use NGS and LC-MS analysis to get abundant information, but analysis among different fermentation time points are not done.”

Response: This suggestion is very important and it has been revised.

  1. Comment: “There are many mistyping and misspelling in the content. Extensive editing of English language required.”

Response: This suggestion is very important and this paper has been edited by a

professional in English.

  1. Comment: “In materials and methods, all the processing steps of the coffee bean after harvest and before analysis should be described in detail.”

Response: This suggestion is very important and it has been revised.

  1. Comment: “Sample codes describe in the content and in the figure legends are different, please unify.”

Response: This suggestion is very important and it has been revised.

  1. Comment: “The legend of Figure 2 should be improved. What's the meaning of inner and outer circles and different colors? And the meaning of the number and percentage beside the circles. The same as Figure 3. What are the meaning of different circles of the same color and different size of circles, And the open circle in upper right side.”

Response: This suggestion is very important and it has been revised.

  1. Comment: “Are DCM and DCnVCs mean the same? If yes, please unify; if not, please check the content and correct.”

Response: This suggestion is very important and it has been revised 

  1. Comment: “Almost no discussion on the results and what are the meaning and importance of your finding.”

Response: Response: This suggestion is very important and it has been revised 

  1. Comment: “Conclusion should be made by summarizing the results briefly, not repeat the results again.”

Response: This suggestion is very important and it has been revised.

  1. Comment: “Extensive editing of English language required.”

Response: This suggestion is very important and this paper has been edited by a

professional in English.

Round 2

Reviewer 1 Report

The manuscript is acceptable in present form.

Reviewer 2 Report

The authors have revised the manuscript according to the reviewers' comments, no more suggestion.

There are still many mistyping exist. Please carefully check all the manuscript.

Some examples.

Line 18, marabolites should be metabolites.

Line 35, which re should be which were.

Line 61, Leuconostoc plantarum should be Lactiplantibacillus plantarum.

Line 131, Basidomycota should be Basidiomycota.

Line 261, Coffee arabica should be Coffea arabica.